# Position: We Need Responsible, Application-Driven (RAD) AI Research

Sarah Hartman [1]  Cheng Soon Ong [1 2]  Julia Powles [3 4]  Petra Kuhnert [1]

## Abstract

This position paper argues that achieving meaningful scientific and societal advances with artificial intelligence (AI) requires a responsible, application-driven approach (RAD) to AI research. As AI is increasingly integrated into society, AI researchers must engage with the specific contexts where AI is being applied. This includes being responsive to ethical and legal considerations, technical and societal constraints, and public discourse. We present the case for RAD-AI to drive research through a three-staged approach: (1) building transdisciplinary teams and people-centred studies; (2) addressing context-specific methods, ethical commitments, assumptions, and metrics; and (3) testing and sustaining efficacy through staged testbeds and a community of practice. We present a vision for the future of application-driven AI research to unlock new value through technically feasible methods that are adaptive to the contextual needs and values of the communities they ultimately serve.

## 1. Introduction

The artificial intelligence (AI) landscape is evolving rapidly, requiring researchers to engage with many aspects of AI integration into society. Innovations span diverse fields, from specific examples in medicine - where AI is helping with early detection of atrial fibrillation and providing real-time health insights in clinical settings (Briganti & Le Moine, 2020), to agriculture - where it promises to better predict crop yield (Elbasi et al., 2022), optimise fertiliser use (Elbasi et al., 2022), and analyse long-term agricultural trends (Hartman et al., 2022). Across both the public and private

[1]CSIRO's Data61, Australia [2]College of Systems and Society, Australian National University, Australia [3]Tech & Policy Lab, Law School, University of Western Australia, Australia [4]Centre for Business Research, University of Cambridge, United Kingdom. Correspondence to: Sarah Hartman <sarah.hartman@data61.csiro.au>.

*Proceedings of the $42^{nd}$ International Conference on Machine Learning*, Vancouver, Canada. PMLR 267, 2025. Copyright 2025 by the author(s).

sectors, the widespread uptake of AI elevates the need for applied AI research to be responsive to contextual needs and societal values (Nissenbaum, 2009), ethical and legal considerations, and public concerns. Further, AI researchers have responsibilities to inform and educate the public regarding actual capabilities and limitations of AI, as an instrumental component of building trust in human-AI collaboration (McGrath et al., 2024).

In recent years, the concept of Responsible AI (RAI) has gained significant traction in industry, government, and academic settings (Adams et al., 2024; Maslej et al., 2024; UNESCO, 2024). The concept of RAI (Confalonieri et al., 2021; Lu et al., 2023) has become more pronounced, as the failures, flaws, and harms of AI have been made visible (Cressie, 2023) due to the embedding of AI into everyday technologies. Responsible AI draws on prior technical work undertaken in the areas of fairness, transparency, and explainability, as means of responding to AI's deficiencies (Confalonieri et al., 2021). Yet, two of the major weaknesses of RAI are that it lacks stable definition and that it fails, at core, to specify who is responsible, for what, to whom (Powles, 2023). Instead, RAI casts responsibility diffusely. Adams et al. (2024, p. 9), for example, define RAI as "the design, development, deployment and governance of AI in a way that respects and protects all human rights and upholds the principles of AI ethics through every stage of the AI lifecycle and value chain. It requires all actors involved in the national AI ecosystem to take responsibility for the human, social and environmental impacts of their decisions".

RAI gestures to earlier frameworks designed to ensure science and innovation are responsive to societal needs, including work on ethical, legal, and social implications, value sensitive design and, most significantly, Responsible Innovation (RI) or Responsible Research and Innovation (RRI). The classic formulation of RI advocates for a deliberative approach to research, centred on four dimensions: anticipation; reflexivity; inclusion; and responsiveness (Owen et al., 2013; Stilgoe et al., 2020). This model has been extensively applied and refined since the early 2010s, and adds considerable sophistication to RAI discourse, particularly around integrating affected communities into the research process (McCrea et al., 2024).

Rolnick et al. (2024) argue for elevating the value of application-driven machine learning (ADML) within the AI research community, positing that ADML research has contributed key advances to the field of machine learning (ML), including by developing tailored methods with wide applicability across ML applications, addressing existing methodological challenges, and diversifying research directions. Building on ADML and RRI, and motivated by RAI's under-specification, **we argue that application-driven AI research will only achieve meaningful scientific and societal impact if it is also accompanied by a responsible approach to research and innovation**. We call this responsible, application-driven AI research (RAD-AI).

The AI research community, with its deep understanding of the technical foundations of AI failures, flaws, and harms, is uniquely positioned to actualise and advance a RAD-AI framework. By directly and systematically addressing public trust and credibility, RAD-AI will contribute to meaningful advances not only in the sectors where AI is applied, but in the policies and processes that calibrate AI deployment to ethical imperatives and the contextual needs of communities.

This paper advocates for people-centred, participatory approaches in RAD-AI research, fostering collaboration, co-design, and efficacy-driven innovation beyond the constraints of purely method-driven approaches. By embedding AI development within real-world contexts and diverse perspectives, we can ensure that these technologies are not only technically robust but also socially and ethically viable. The question is: how do we get there?

## 2. From AI to RAD-AI: Framing AI Research as a Sociotechnical Challenge

### 2.1. Paradigms: From Technological Determinism to Centring People

The AI community needs to be wary of adopting the paradigm of technological determinism: the theory that technology inherently and autonomously changes society, unaffected by social pressures and constructs (Kline, 2015). Technological determinism can lead to AI researchers assuming that their work is inherently desirable, with the affordances of technology "becom[ing] ends in themselves, instead of the means for creating desired social ends" (Kline, 2015, p. 109). Instead of assuming, researchers should ask critical questions to ensure that their work does indeed align with contextual needs and societal values, shaping how technology functions, for whom, and to what effect (Davis, 2020). As Stilgoe and Cohen (2021) argue, by its nature, technological determinism precludes co-creation between those developing technologies and affected communities, since communities are the assumed eventual adopters of

technology, rather than active players in how it is researched, developed, and shaped (Stilgoe & Cohen, 2021). The paradigm sees communities as "onlookers... displace[d] from the action" (Zenz & Powles, 2024, p. 3), incorrectly assuming that adoption is inevitable and public approval is not required (Stilgoe & Cohen, 2021; Zenz & Powles, 2024).

Technological determinism can be compared to an abandonment of experimental design in statistics (Fisher, 1971). Before the advent of powerful computing, experimental design was a fundamental component of applied research, shaping both the feasibility and cost-effectiveness of studies. It ensured that resources were used efficiently to achieve statistically significant results. Just as rigorous experimental design once safeguarded research integrity, in AI development we must adopt a RAD-AI lens to ensure AI systems are thoughtfully designed.

### 2.2. Breaking Out of AI (Black) Boxes

RAD-AI calls for breaking free from the rigid structures that confine AI research - structures shaped by disciplinary boundaries, implicit assumptions during problem definition, narrow success metrics, and the opacity of AI's black box methods. These constraints manifest most critically in how researchers *frame problems* and *select ML methods*, shaping the trajectory of AI development. When these stages are constrained, researchers may struggle to develop solutions that are truly aligned with real-world complexities and needs. We explore both of these challenges below.

**Problem Framing.** One challenge we identify is the boundaries that experts impose when defining problems. Traditionally, professionals such as doctors, engineers, lawyers, and computer scientists, are taught to bound, map, characterise, and document a problem, situating it in a specific technical framing (Li, 2007). To distil problems into technical solutions, the socio-political complexities that define them are mostly glossed over, creating a skewed and incomplete approach to knowledge and in some cases even a "sidelining much of the data so painstakingly collected" (Li, 2007, p. 126). While expertise is important, it may narrow the frame of reference and leave out critical components of the complex world. What scientists initially hypothesise, collect data on, and build models around reflects existing power dynamics and interests, worldviews, and technical training, potentially missing critical nuances. We may miss understanding and nuance because it is not captured in the original framing of the question, the data collected, or the type of result being prioritised, including the specific metrics used to measure success (Rolnick et al., 2024). Addressing real-world problems with algorithms and AI systems requires looking beyond the constraints experts impose - both in how challenges are framed and in the broader contexts

that shape them. These contexts, spanning history, politics, and power (Powles, 2023), not only define but also limit potential solutions.

A second challenge is where the expert box is also bound by scientific reductionism – the scientist's proclivity for oversimplification of reality (Greenhalgh, 2025). Concerns about oversimplification of AI approaches are evident across various fields, ranging from medicine (Sturmberg & Mercuri, 2024; Greenhalgh, 2025) to agriculture (Tzachor et al., 2022). For example, model emulation approaches (Gladish et al., 2018; Bolt et al., 2023; MacKinlay et al., 2025), such as those applied to complex agricultural models (Powell et al., 2023) offer crucial speedups and simplifications, effectively 'jailbreaking' their complexity. However, scepticism toward AI innovations often stems from the notion that AI seeks to replace, rather than complement and build off, well-established agricultural models. This kind of reductionism - which can involve excessive as well as insufficient parametrisation - generally overlooks the nuanced, interconnected, interdependent, and evolving nature of problems. It can also hide assumptions. As Sturmberg and Mercuri (2024, p. 3) note, "no problem exists in isolation" rather "all problems are embedded within a unique context." Similarly, RAD-AI approaches will have nuanced differences depending on the context in which they are applied.

**Method Selection.** A key challenge in AI method selection is the lack of interoperability of complex models. While ML approaches such as neural networks, support vector machines, random forests, and k-nearest neighbours can be tuned for high accuracy, they often obscure intermediate decisions, making it difficult to understand how conclusions are reached (Loyola-Gonzalez, 2019; Confalonieri et al., 2021; Mahinpei et al., 2021). These black-box models embed hidden assumptions and biases (Suresh & Guttag, 2021), adding to the oversimplifications that arise during problem framing. The challenge is further amplified when AI models are embedded within larger, interconnected systems, where multiple models interact in ways that reduce transparency and make it harder to diagnose errors or unintended consequences. One response is to shift towards 'white-box models'– such as decisions trees (Breiman et al., 1984) and rule-based systems (Loyola-Gonzalez, 2019), which offer greater interpretability. However, limiting RAD-AI research to white-box approaches risks overlooking valuable insights that could emerge from engaging critically with black-box methods and addressing their interpretability challenges in applied contexts.

These constraints pose significant challenges in how AI research is conducted, shaping not only the problems AI researchers tackle but also how they approach solutions.

## 2.3. Law, Policy, and Public Discourse

With AI exploding from a niche scientific sub-discipline to an intense focal point of state policymaking and public discourse, the AI research community must reimagine what it means to conduct and communicate science in areas that necessitate societal trust. This starts with legal requirements - first in relation to AI, and second in relation to the application domain.

An increasing number of legal and policy developments are focusing on AI, with AI explicitly mentioned in laws passed by over 30 countries by 2024 (Maslej et al., 2024; UNESCO, 2024). Three main motivations are driving AI regulation: to "address a public problem"; "protect, respect or promote fundamental and collective rights"; and create the conditions to "achieve a desirable future" (UNESCO, 2024, p. 41). Particular attention has been given to AI applications in areas that are considered "high risk", including education, health, policing, and the workplace. Yet unlike traditional policymaking, tech policy tends to privilege a multi-stakeholder approach - directly involving the tech industry and its concerns in scoping regulation – with more marginal attention to social, economic, and environmental externalities (UNESCO, 2024).

Engaging with law and policymaking processes and associated public debates provides researchers with opportunities to identify priorities for their own research and public engagement, often with great specificity (e.g., concern over face recognition, data extraction, and high risk AI applications), rather than lofty principles and goals (Schiff et al., 2021). This should focus not only on AI-specific laws and policies, but particularly on established (and often extensive) legal requirements in the given application domain.

Understanding national priorities and strategies around AI and application domains will be beneficial for understanding policymaking agendas and for researchers who rely on funding from national science agencies or foundations. RAD-AI provides a competitive edge in this context, as funding bodies increasingly stipulate both transdisciplinary and engaged research as component of funding (Bednarek et al., 2025). The push to safeguard individual and community agency, along with building domestic digital capability and sovereignty, is already shaping regulation and funding globally (Chahal et al., 2022; NSF, 2023; Mügge, 2024), and must not be overlooked by the AI community.

Maintaining public approval (often termed 'social license') is essential to the effective conduct of research. Communities generally expect a very high safety bar for new technologies, presenting a challenge for AI applications that require broad real-world testing to achieve market readiness (Hemesath & Tepe, 2023). For example, public apprehension has hindered the testing and deployment of autonomous

vehicles (Stilgoe & Cohen, 2021; Hemesath & Tepe, 2023). Yet in application domains from food to pharmaceuticals to gene technology, both pre- and post-market testing are non-negotiable - there is no reason why applied AI should be any different.

Strengthening the "connection between [AI's] capability improvements and AI's social or economic impacts" is critical (Narayanan & Kapoor, 2024). While method-centric studies will prioritise a "narrow set of evaluation metrics" such as test loss and accuracy (Rolnick et al., 2024, p. 2), they may overlook an innovation's feasibility in solving real-world problems or its alignment with the societal values and priorities of the communities it serves. The "bottleneck for impact" often lies in the pace of product development and the rate of adoption, rather than methods advancement (Narayanan & Kapoor, 2024). Addressing these challenges requires deliberate efforts to align technical progress with societal needs, fostering both innovation and public trust.

### 2.4. Method-Driven Research Cannot Account for Sectoral Nuances

A strong argument for ADML centres around creating algorithms and systems that address real-world challenges (Rolnick et al., 2024). To do this, these algorithms and systems must engage with pre-existing sector or problem-specific considerations. In these intersecting, complex systems, AI solutions do not exist in a vacuum but interact with sectoral nuances, including ethical considerations, in ways that must be considered when developing RAD-AI. Engaging with the physical and social milieu of a context is nuanced and complex, yet simultaneously stimulating for researchers and essential for affected communities (McMillan-Major et al., 2024). Early consideration of the context to which AI is being applied facilitates identifying and addressing undesirable outcomes from this complex and often messy process of engaging with real-world challenges.

**Nuances in Ethics.** Applying AI to new fields of science carries considerable excitement. The 2024 Nobel Prize awards highlighted this transformative potential, showcasing AI advancements in the disciplines of Physics and Chemistry (Li & Gilbert, 2024). However, as AI's legitimacy and impact expand beyond computer science, the field is also interacting with disciplines that have strong histories of ethical standards, such as the Hippocratic Oath for physicians and the Obligation of the Engineer for engineers. These ethical standards guide the everyday workings of these professions and their fundamental societal obligations.

As the AI research community actively discusses what its ethical standards should be, and learns from the ethical codes of application domains, it can draw inspiration from existing sectors with established ethical commitments.

It is important to recognise that effective engagement with nuances in ethics at various levels will be messy. The integration of many fields has the potential to lead to deprioritisation or event displacement of existing priorities and ethics concerns. This has been seen in the Water, Sanitation, and Hygiene (WASH) field, where alignment with global health and development paradigms has obscured intrinsic aspects of the sector (de Wit et al., 2024). For example, prevailing paradigms prioritise universalisation ("creating global, mobile models"), responsibilisation ("locating responsibility with communities and individuals"), technicalisation ("rendering WASH actionable as a technical, depoliticised, global project"), and metricisation ("making problems and solutions measurable") while overlooking localised values and the complex reality of lived experience in poverty (de Wit et al., 2024, p. 5). Recent work indicates that in global water-climate contexts, the growing emphasis on climate-aligned WASH priorities has sidelined historic WASH priorities of equity and universal access to WASH (Cullen et al., 2025). Similarly, in agriculture, applying ADML without intentional design could exacerbate historical and ongoing ethical issues, such as the use of child labour and the dispossession of smallholder or indigenous agricultural land (Li, 2007; Hartman et al., 2022; Tzachor et al., 2022).

Recent work towards helping the AI community consider responsible practices in their methods development has highlighted that while high-level ethics principles exist, operationalising them remains a challenge (Sanderson et al., 2024). They highlight that ethical aspects can interact with AI systems in ways that create trade-offs between accuracy and explainability. While AI developers may be aware of this, Sanderson et al. (2024) explain that it does not follow that they would be willing to engage and prioritise accuracy and performance over ethical considerations, proposing guidelines for managing these tradeoffs.

These examples underscore the importance of centring domain-specific ethical considerations when developing AI to ensure that its benefits do not come at the expense of vulnerable populations or established values. Further, they foreshadow RAD-AI's complexities, while demonstrating the rich opportunities for advancing AI methods through an applied lens.

**Nuances in Needs.** AI researchers have the opportunity to engage with the unique needs of sectors, which are shaped by complex histories and challenges, and are also rich with opportunities. Increasingly, sectors are now outlining their specific needs—offering roadmaps to guide the AI community on what domain experts need.

In agriculture, AI is increasingly seen as a component of an integrated suite of sociotechnical interventions needed to address food security. For example, global food production data, fundamental to tracking and responding to food secu-

rity issues, is riddled with social and technical challenges that result in data scarcity worldwide (Kebede et al., 2024). Domain experts have suggested addressing this challenge in a nuanced way that involves thoughtfully integrating AI into sociotechnical interventions (Kebede et al., 2024). In this sector, simply applying AI will not result in the integrated solutions that Kebede et al. (2024) argue will address the problem.

In the geosciences, calls for tailored AI ethics centre on the needs specific to the geosciences – particularly as the field evolves towards multidisciplinary "social geosciences." (Cleverley, 2024). Domain-specific needs include avoiding hyperbole when predicting natural disasters and the spreading of obsolete, inaccurate misinformation (Cleverley, 2024).

Lastly, in public broadcasting, the pre-existing need to deliver innovative content while upholding transparency, maintaining human oversight, and safeguarding reputation is not just optional, but obligatory. The need to incorporate sector-specific benchmarks has led to pioneering research that uses an open, extensible framework to align large language models with existing AI principles for broadcasting (Seneque et al., 2024). Specifically, in this space, the aim is to ensure the trademarks of journalism – human oversight and editorial expertise – are maintained while still leveraging the potential of AI.

These examples point to the opportunity RAD-AI opens to work across sectors and with many types of actors to address contextual needs, responsibilities, and requirements.

## 3. A Three-staged Approach to RAD-AI

The integration of Responsible Research and Innovation principles into application-driven contexts requires a deliberate, structured approach. Below, we propose a three-staged pathway as a starting point to guide the AI community in breaking out of the box to achieve RAD-AI in practice (Fig. 1). This pathway consists of laying a strong foundation, navigating complexity, and testing and sustaining efficacy to ensure that AI solutions are effective, ethical, and responsible. It serves as a complement to existing resources such as the American Association for the Advancement of Science (AAAS) Decision Tree for the Responsible Application of AI (AAAS, n.d.), which provides a set of questions using a decision tree format to assist practitioners with identifying whether to develop or deploy AI solutions. It should also be considered alongside (or incorporated into) relevant international standards including the one on AI management systems (International Organization for Standardization, 2023).

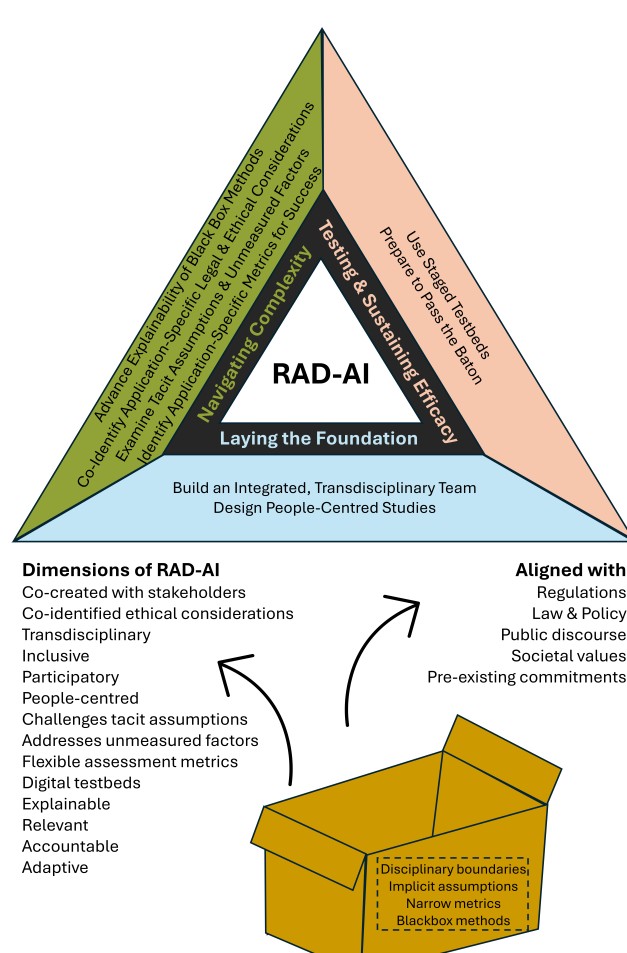

*Figure 1.* A three-staged framework for RAD-AI and visualisation of the dimensions and alignments that break from AI's black box.

### 3.1. Laying the Foundation

Intentionally planning RAD-AI from the onset of a project lays the foundation for success. This stage focuses on building an integrated, transdisciplinary team and people-centred studies.

**Build an Integrated, Transdisciplinary Team**. An interdisciplinary – even transdisciplinary (i.e. the inclusion of non-traditional academic knowledge holders) – team from the start makes accomplishing RAD-AI more feasible. By drawing on the expertise of individuals from diverse disciplines, world views, and lived experiences, the team can harness a broader range of skills, perspectives, and creativity and break out of the problem framing box. Recent calls in the AI community have focused on integrating AI with the humanities and social sciences in order to develop greater legitimacy, credibility, meaningfulness, and capabil-

ity (Chades et al., 2025). Historically, incorporating domain knowledge was crucial in the success of expert systems, one of the pioneering methods of ADML systems (Confalonieri et al., 2021). Domain knowledge was the key that allowed ADML "to reason, draw new conclusions, and to generate explanations" (Confalonieri et al., 2021, p. 14), legitimising its value. Its importance and success reiterate the centrality a plurality of domain expertise plays in methods development and points at the need to continue innovating on how domain knowledge is incorporated.

To advance responsible, practical black-box models, Mahinpei et al. (2021) highlight the need for better collaboration with experts to identify new, relevant concepts to focus on in intermediary quality assessments and ensure ultimate model outputs are interpretable to end users. Co-creating models with applied, transdisciplinary experts ensures practitioner-interpretable, actionable, and pragmatic outputs. Incorporating checkpoints with transdisciplinary experts also helps combat the oversimplification (i.e. problem framing box) that often occurs in AI.

**Build People-Centred Studies.** This step is targeted at developing transparency, trust, and interpretability. The benefits of inclusive, participatory, and people-centred research are well-recognised (London et al., 2020; McGrath et al., 2024; UNESCO, 2024; Adams et al., 2024; Bednarek et al., 2025). Two popular styles of people-centred research are human-centred design and community-based participatory research. The former is an iterative approach to aligning human desires with technologically feasible solutions, while the latter revolves around intentional, long-term relationship-building between researchers and the community (Chen et al., 2020). Both uphold values of co-creation, engagement with relevant stakeholders, and are adaptable and allow for two-way knowledge exchange (Chen et al., 2020). Depending on context, one style or a combination of the two may be more appropriate (Chen et al., 2020).

At its best, consultation or co-creation unlocks a means of distilling the complexities of politics, history, and power (Li, 2008), and allowing for context-sensitive and equitable solutions (Suresh et al., 2022). For example, this approach makes space for diverse ways of knowing in research, such as indigenous knowledge and value systems, which have been identified as a major gap in the current global RAI ecosystem (Adams et al., 2024).

Immense value to the research process comes from allowing the community "to constructively challenge, co-create, or innovate" (Stilgoe & Cohen, 2021, p. 850) in the research itself. Early engagement improves the explainability of models, since "there is a gap between explainability and the goal of transparency, since explanations primarily serve 'internal' [computer science] stakeholders rather than 'external' [community] ones." (Confalonieri et al., 2021, p.

15). Further, AI's growing ability to autonomously reach conclusions from data and provide reasonable theoretical justifications for them (Novy-Marx & Velikov, 2025) makes human oversight imperative to ensuring ethical considerations and preserving complexity of real-world systems. Hallucinations also pose a risk in these applied contexts, highlighting an ever increasing need for responsible methods development (e.g. since even small tweaks in prompting can produce different narratives and theoretical rationales from AI) alongside improvements in large language models to be more context-aware and evidence-based (Novy-Marx & Velikov, 2025).

Finally, achieving this step requires continuous, adaptive participatory engagement that evolves over time, ensuring that expectations and relationships remain resilient and aligned with the project's goals (London et al., 2020). London et al. (p. 1) argue that "what makes for good community engagement is not simply the extent but the fit or alignment between the intended approach and the various contexts shaping the research projects", which can include "the scale and scope of the ... [issue], the capacities and resources of the researchers and community leaders, and the influences of the sociopolitical environment".

### 3.2. Navigating Complexity

It is a trained tactic of the AI researcher to draw a box around the exact, incremental AI problem a study is being designed to advance. To break out of that box, this step focuses on ensuring relevance and accountability by advancing explainability, co-identifying ethical considerations, challenging tacit assumptions, and addressing unmeasured factors.

**Advance Explainability of Black Box Methods.** While the boxes that constrain the AI community may persist, RAD-AI should advance methods that address public concerns. For black-box methods, one way to do this is by increasing explainability of global methods, local methods, or introspective methods (Confalonieri et al., 2021). Recent work has shown promise towards this end by optimising submodels of complex systems (Chen et al., 2024) and making intermediate steps more interpretable for humans by employing intermediate concept learning to convert black-box model processes into comprehensible information (Mahinpei et al., 2021). Also, differentiable models show great promise in addressing the black-box nature of AI and leveraging the advantages of both physical and AI models (Shen et al., 2023). Lastly, combining white- and black-box models shows potential (Loyola-Gonzalez, 2019). However, more innovation is needed to improve methods to achieve RAD-AI, since such emerging methods reveal multiple new methodological needs for the AI community to explore. These include information leakage and unintended, misleading predictions, where extra information is encoded

between steps beyond the intended concepts, indicating that some of the concepts may not be relevant for the model task (Mahinpei et al., 2021). In the case of differentiable models, their use raises new technical challenges of memory usage and vanishing gradients (Shen et al., 2023). Lastly, RAD-AI will require advancing skills in prompt engineering (Novy-Marx & Velikov, 2025).

**Co-Identify Application-Specific Legal and Ethical Considerations.** RAD-AI research reframes research projects in a way cognisant of the context and nuances of legal and ethical considerations in the specific application and sector. Co-identifying these considerations involves asking 'what is the history, context, and underlying (system-level) drivers or constrainers of the problem?' and engaging with pre-existing law, policy, ethical codes and commitments, and public discourse. A pre-requisite for developing and assessing RAD-AI is having established principles (e.g. in the work of Seneque et al. (2024) that advances methods for aligning large language model outputs with safety and accuracy as outlined in relevant organisational AI principles). If ethical principles do not exist, these should be established here. This step also focuses on interrogating the intersection of legal and ethical considerations from different domains, as well as recognising different ethical priorities and interpretations across public, private, and non-profit sectors. Some sectors may emphasise or omit certain ethical concerns; for example, NGO and public sector documents are more likely to engage in participatory processes (Schiff et al., 2021).

**Examine Tacit Assumptions and Unmeasured Factors.** Every study is built on tacit assumptions and unmeasured factors that influence the outcomes. Studies have been shown to reflect "the point of view that the modeler decides to capture" (Confalonieri et al., 2021, p. 14), which can shape the research in ways that may not be immediately apparent. By critically examining the tacit assumptions and unmeasured factors that are brought to the study, research can build transparency (Sturmberg & Mercuri, 2024) and reflect a more comprehensive understanding of the context. For example, while a technological determinist may assume AI is both inevitable and desirable, this may not always be the case. In fact, Powles (2018) and the AAAS decision tree (n.d.) prompt the practitioner to reflect on whether AI is the right technological solution before the research begins. In this phase, researchers should ask 'what am I assuming about the context, in terms of the way I am framing it and the dimensions I am capturing and omitting? What is being left unmeasured or unaddressed?'

**Identify Application-Specific Metrics for Success.** Once working in a RAD-AI setting, the need for tailored benchmarks and metrics for success in addition to standard benchmarks is paramount for evaluating efficacy in a complex,

localised context (Rolnick et al., 2024; Seneque et al., 2024). Application-specific metrics for success are the relevant tests and benchmarks that can be used to appropriately assess the usability and validity of the model outputs for the intended context (Loyola-Gonzalez, 2019; Shen et al., 2023; Rolnick et al., 2024). This could include interpretable, intermediary model quality assessments and interpretable model outputs for end users (Mahinpei et al., 2021). Alternatively, Loyola-Gonzalez suggests using the Delphi method as "a statistical method for validating the suitability of an applied machine learning model by using the opinions of several experts in the application area" (2019, p. 154108). Other fields, such as finance, are already calling for changes in their evaluation standards in light of AI-enabled advancements (Novy-Marx & Velikov, 2025). Establishing context-specific benchmarks increases the chances that AI models are aligned with the needs of the specific application.

### 3.3. Testing and Sustaining Efficacy

The final stage of RAD-AI focuses on scaling and preparing both AI innovation and the AI research community for sustainable efficacy.

**Use Staged Testbeds.** Staged, digital testbeds provide flexible, adaptable environments for testing AI systems over time, allowing researchers to identify potential hazards, and respond to changes and evolving requirements (Tzachor et al., 2022). These testbeds are an essential tool for ensuring that AI models are scalable, sensible, and responsible (Sumpter, 2024). As UNESCO (2024, p. 45) notes, "testbeds should be part of the 'agile' regulatory toolbox available to policymakers to accommodate the fast pace of technological innovation." By using digital testbeds, researchers can iteratively refine their models, testing them against both technical and ethical standards while evaluating their robustness in real-world contexts to ensure the results make sense in a complex world with both stated and implicit contexts. Importantly, this stage should also consider how the model and underlying training data might change with time and space. It asks 'how scalable is this methodology? How are the ethics considerations, unmeasured factors, and tacit assumptions identified previously manifesting at scale? Will this research age well, and what will be its legacy in our grandchildren's generation?'

**Prepare to Pass the Baton.** Drawing from the leadership field, thought-leaders in the AI community have an opportunity to critically reflect on the current field and shape the legacy it will leave for the next generation. Leadership has the ability to "create, uphold, disrupt and recreate systems" (Taylor & Manning-Ouellette, 2024, p. 50); the values and ways of thinking leaders pass on are the 'baton' that future leaders start with. The question is, what is the 'baton' that is being passed on, and how well is the next generation

equipped to carry on the legacy?

When we consider values, do we actively engage with the concerns shaping public acceptance of AI? Are we maintaining public trust in science by conducting our work with intellectual humility - the recognition of the limits of one's knowledge (Koetke et al., 2025)? Intellectual humility manifests in many ways including acknowledging individual limitations and gaps in knowledge, actively listening to others - including the public - and reevaluating assumptions as new information emerges (Koetke et al., 2025). Testbeds provide a dual benefit here as they are a proving ground of the 'baton' while also offering a place to assess, refine, and iterate on what is being passed forward. They help evaluate how the next generation has been prepared, identifying both strengths and areas of improvement. Importantly, "by centering systems thinking,. . . leaders can pass a baton of innovative and equitable systems to the next generation" (Taylor & Manning-Ouellette, 2024, p. 2).

Leaders should actively mentor, educate, and co-create throughout the project to ensure that the project's legacy extends beyond the original baton-holder. This calls for current leaders to build a RAD-AI community of practice that fosters collaboration and knowledge sharing. Such communities are important for cultivating the skills and dynamic practices needed to empower researchers to thoughtfully develop technologies while meaningfully addressing ethical concerns (McMillan-Major et al., 2024). Moving forward, there is an opportunity for leaders to develop a RAD-AI community of practice that ensures that future leaders are equipped to sustainably advance the field.

## 4. Alternative Views

Counter-positions to RAD-AI are: (1) research in artificial general intelligence (AGI), not contextual AI, is needed; (2) a checklist, not adaptive, participatory processes, is the real necessity; and (3) 'open' RAI is sufficient for community participation, inclusion, and building trust.

**AGI will address our needs**. While RAD-AI will allow for engaged, contextually-embedded research, it is not generalisable in the way AGI and its precursors aspire to be. Our paper refers to AGI in its broadest sense, defined in the 2025 AI Safety Report as "Potential future AI that equals or surpasses human performance on all or almost all cognitive tasks" (Bengio et al., 2025, p. 218). While this scale of intelligence is yet to be achieved, even today's most advanced AI models have concerning established and emerging risks that the most advanced mitigation techniques have not been able to overcome (Bengio et al., 2025). In particular, the gargantuan scale of the AGI learning process is near impossible to understand and obscures failure points, as learning gives rise to emergent, intermediary, and complex patterns

from raw inputs. AGI also develops unforeseen capabilities (Thieme et al., 2023). This method, while powerful, often leads to a loss of explainability, making it difficult to understand how AGI systems arrive at their conclusions. Additionally, the reuse of a few foundational models in applications homogenises any defects of the original models and embeds the defects in any downstream models (Thieme et al., 2023). Rich Sutton's "The Bitter Lesson" argues that approaches leveraging massive computation, rather than human knowledge or domain expertise, have historically yielded the most effective AI systems (2019). This framing tends to sideline critical contextual considerations of RAD-AI, reinforcing a technological deterministic viewpoint that AI can, and should, develop independently of human-driven constraints, including social, ethical, and domain-specific factors. RAD-AI, through its three-staged approach, dynamically aims to minimise such concerns in AI development.

**It's time for an RAI checklist**. At the other end of the spectrum, the more operationally-minded may propose implementing RAD-AI through a prescribed RAI checklist or RAI metrics (similar to (OECD.AI, n.d.)). While it may be attractive to offer 'one-and-done' checklists, metrics, assessments, and tools, the danger is in execution: a checklist by its nature oversimplifies and reduces the nuances of context. Contrastingly, the RAD-AI framework, as well as the AAAS decision tree (n.d.) have an important distinction from a checklist - they treat the process of engaging with RAI as a continuing, dynamic *conversation*. The complexities of AI, including its influence across various dimensions such as societal issues, ethics, and domain-specific challenges, make the pursuit of a single defining metric impractical. As fluid as public discourse and societal concerns are, so too must be the formats of engaging with RAD-AI.

**'Open' AI is all you need**. Following early efforts by Sonnenburg et al. (2007), open source has gained strong support in the research community to democratise AI innovations through platforms like GitHub. Open Source allows for transparency, reusability, and collaboration around innovations in AI, allowing methodological researchers and applied scientists to easily access, extend, and apply methods to new problems. While it may seem sufficient for RAI practices, 'open' AI "often lack[s] precision", omits "scrutiny of substantial industry concentration", and "often incorrectly appl[ies] understandings of 'open' imported from free and open-source software to AI systems" (Widder et al., 2024, p. 828). These challenges conflict with our proposed three-staged approach that targets RAD-AI practices. Though open-source has its place, it cannot fully address RAD-AI needs.

# 5. Conclusion

In this position piece, we identify the need to integrate Responsible Research and Innovation and Responsible AI techniques with application-driven AI research (RAD-AI). We propose a three-stage process that: (1) brings together transdisciplinary teams for people-centred studies; (2) tailors methods, ethics, assumptions, and success metrics to context; and (3) builds for efficacy through iterative testing and future planning. The benefits of RAD-AI include building a civically-engaged research community, enhancing public confidence in AI and the data and models on which it depends, and providing mechanisms for better resourced research. The RAD-AI approach does this by breaking down the 'boxes' of disciplinary boundaries, problem definition, narrow success metrics, and opaque processing, that currently limit the AI community.

The most important and challenging ongoing work of RAD-AI will involve applying its transdisciplinary and participatory methods to addressing the systemic needs of marginalised and underserved communities. As this involves engaging with systemic disadvantage, researchers will be tested by some of the most difficult and rewarding challenges in AI research and innovation. As the AI field continues to grow, we predict that those who engage with RAD-AI will not only drive innovation but will also shape the future of AI, ensuring it is highly adaptive and responsive to contextual needs and societal values.

The time is ripe for AI research that truly contributes to a better shared future. It's time to make AI research RAD!

## Acknowledgements

We thank the reviewers and our colleagues at CSIRO, especially from the Collaborative Intelligence (CINTEL) Future Science Platform (Cecile Paris), Data61 (Cara Stitzlein, Dan Pagendam, Andrea Powell, and Conrad Sanderson) and Agriculture and Food (Roger Lawes), for their insight and support throughout this work. SH was supported by a CSIRO CERC Postdoctoral Fellowship.

## Impact Statement

This paper presents work whose goal is to advance the field of Machine Learning in a way that contributes to a better shared future by ensuring AI research is adaptive and responsive to contextual needs and societal values.

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
