# OpenReview forum: "Position: We Need Responsible, Application-Driven (RAD) AI Research"
_ICML.cc/2025/Position_Paper_Track — ICML 2025 Position Paper Track poster_

### Official Review · Reviewer_kC5K · 2025-03-09

**Significance:** 3
**Argument Clarity:** 3
**Rating:** 4
**Confidence:** 3

**Questions:**

.How would you respond to the worry that your proposal somewhat lacks distinctiveness? (I don't consider this a major concern, but I would like to hear your thoughts on this.)
How would you address the worry that the framing of the "AGI" alternative view section is lacking? (I consider this a minor problem, but one that should be addressed.)

Thank you for this helpful contribution.

**Discussion Potential:**

2

**Paper Summary:**

The paper advocates for a shift toward Responsible, Application-Driven (RAD) AI research. It begins by arguing that responsible AI is under-specified in crucial ways. It builds on insights from “application-driven machine learning (ADML)” and “Responsible Research and Innovation (RRI)” to correct this under-specification. The account is clear, well motivated, and feels helpful. I enjoyed reading this paper. Thank you for your contribution.

**Position:**

Yes

**Position In Title:**

Yes

**Related Work:**

4

**Strengths And Weaknesses:**

Strengths:
* Thorough Background Analysis: The paper provides a comprehensive and up-to-date analysis of the background underpinning its proposals.
* Clear Framework Articulation: It clearly outlines the motivation, aims, and components of the RAD-AI framework.
* Strong Justification of Principles: The high-level principles are robustly defended throughout the paper.

Weaknesses
* (Minor: Distinctiveness): The principles proposed are very high level, which is appropriate for a short position paper. This also makes it slightly harder to appreciate the proposal’s distinctiveness. I similarly wonder about how much discussion potential the paper has. Nevertheless, the paper feels helpful to me, and the positive proposal is clear and well motivated. So I do not consider this to be a serious weakness.
* (Minor: AGI “alternative view” discussion) This section is appropriately concise but in its current form requires improvement. (Note that I consider this easy to fix. So I do not weigh this limitation strongly.)  (1) It does not state what these authors mean by “AGI”, which is a notoriously contested and often ill-defined concept. (2) It presents assumptions as fact: such as, “AGI deliberately avoids the nuances of context”. At the bare minimum, authors should be extremely specific about what they mean by “AGI”. Personally, I recommend shifting the focus to something less ambiguous. For example, consider instead contrasting RAD-AI with the view that “scaling is all we need”.

**Support:**

4

---

> ### Author Rebuttal · Authors · 2025-04-01
>
> Thank you for your thoughtful feedback on our paper. We appreciate the generous encouragement and suggestions to make the position stronger, and are enthused to see the ideas resonating. In this response we elaborate further on the distinctiveness of the proposal and welcome your feedback on the components addressing AGI.
>
> 1. Distinctiveness of the Proposal | We are grateful that the RAD-AI framework has a strong element of familiarity to experts in the field. If challenged about distinctiveness, we would identify that there is very little work in the crowded Responsible AI field that focuses in detail on contextual applications. We strengthen our case for RAD-AI by elaborating on examples from the paper and introducing new ones (see responses to Reviewers 1 and 2). We imagine that this will also help generate further discussion, particularly if these examples cut in both positive and negative directions. Through these case studies, we aim to demonstrate that while the RAD-AI framework is built on high-level concepts, it truly comes into its own when it deals with specific contexts.
>
> 2. Further Elaboration on AGI | Thank you for pointing out the gaps in this section. We gladly take the opportunity to address them here, and will update the paper if accepted.
>
> We acknowledge the reviewer's concern about the lack of precision about the phrase AGI, but prefer to continue using it, and will follow the glossary of the recent AI Safety report (Bengio et. al. 2025) as a definition. However, we agree with the reviewer that we should clarify the baggage involved in using the phrase AGI, that implies a scaling is all you need approach, as discussed in Rich Sutton's "The Bitter Lesson". Therefore, we propose to replace the text:
>
> "While RAD-AI will allow for engaged, contextually-embedded research, it is not generalisable in the way AGI and its precursors aspire to be. With goals to understand, learn, and apply humanlike intelligence across tasks or contexts, AGI juxtaposes RAD-AI. AGI deliberately avoids the nuances of context, while eliminating the dynamism of exchange across communities that is at the heart of the RAD-AI framework."
>
> with the following text:
>
> "While RAD-AI will allow for engaged, contextually-embedded research, it is not generalisable in the way AGI and its precursors aspire to be. This paper refers to AGI in its broadest sense, defined in the 2025 AI Safety Report as "Potential future AI that equals or surpasses human performance on all or almost all cognitive tasks" (Bengio et al. 2025 Pg 218). While this scale of intelligence
> is yet to be achieved, even today's most advanced AI models have concerning established and emerging risks that the most advanced mitigation techniques have not been able to overcome (Bengio et al. 2025). In particular...[continues with original text until the end of sentence in line 412].
>
> Rich Sutton's "The Bitter Lesson" argues that approaches leveraging massive computation, rather than human knowledge or domain expertise, have historically yielded the most effective AI systems (2019). This framing tends to sideline critical contextual considerations of RAD-AI, reinforcing a technological deterministic viewpoint that AI can, and should, develop independently of human-driven constraints, including social, ethical, and domain-specific factors. RAD-AI, though its three-staged approach, dynamically aims to minimise such concerns in AI development."
>
> As an aside to the reviewer, lastly, RAD-AI provides a path to address highly specialised fields like Insurance or Education, where understanding nuance is essential. While RAD-AI may not promise the fastest route to AI methods development, the key to successful human adoption lies in the details. RAD-AI focuses on developing solutions that are fit-for-purpose, ensuring that AI applications are tailored to meet specific needs. This approach not only enhances the effectiveness of AI but also aligns with the goal of creating responsible and context-sensitive AI systems. By prioritising context-specific applications, RAD-AI ensures that AI solutions are both practical and beneficial for the intended users. We advocate for RAD-AI as a complementary and necessary alternative to AGI that in turn, will bring diverse, rich approaches to the ML community broadly.

---

> > ### Comment · Reviewer_kC5K · 2025-04-07
> >
> > Thank you for your reply and for this helpful contribution.
> >
> > Your proposed revisions for the AGI subsection sound appropriate to me.
> >
> > (I am maintaining my current score since my score reflected the fact that I considered this issue very easily remedied.)

---

### Official Review · Reviewer_AWzq · 2025-03-17

**Significance:** 2
**Argument Clarity:** 3
**Rating:** 3
**Confidence:** 4

**Questions:**

- What is the main call-for-action of this paper? I mean, what is the expected impact of your paper on the AI research community?

**Discussion Potential:**

2

**Paper Summary:**

This position paper advocates for a Responsible, Application-Driven AI (RAD-AI) approach, arguing that meaningful AI research must integrate ethical, contextual, and application-specific considerations. It builds on prior concepts such as Responsible AI (RAI) and Responsible Research and Innovation (RRI), but it critiques existing frameworks for their lack of specificity regarding accountability and effectiveness.

The paper introduces a three-staged RAD-AI framework to guide AI research toward responsible and context-aware implementation: Laying the Foundation - encourages forming transdisciplinary teams and conducting people-centered studies; Navigating Complexity – Calls for advancing explainability, challenging assumptions, and aligning AI research with sector-specific ethical, legal, and policy considerations; Testing and Sustaining Efficacy – Recommends staged testbeds for evaluation and fostering a community of practice for long-term sustainability.

The paper strongly rejects the notion of AI research as purely technical and instead promotes application-driven AI that is deeply embedded in real-world contexts. It argues that AI research should be transdisciplinary, participatory, and ethically grounded, addressing specific societal needs rather than being solely motivated by technical advancements.

**Position:**

Yes

**Position In Title:**

Yes

**Related Work:**

3

**Strengths And Weaknesses:**

Strengths:
- The three-stage RAD-AI framework (foundation, complexity navigation, and sustainability) provides a concrete, structured path for researchers and practitioners
- Overall, the claims provided by the authors sound very reasonable

Weaknesses:
- While the paper effectively argues for RAD-AI, it lacks empirical validation or detailed case studies.
- As a technical AI researcher which is also involved in Responsible AI discussions, the framework and the claims sound obvious to me.

**Support:**

2

---

> ### Author Rebuttal · Authors · 2025-04-01
>
> Thank you for your thoughtful feedback on our paper. We appreciate the opportunity to elaborate further through case studies and to clarify the main call-to-action, the distinctiveness of the proposal, and the expected impact of our work on the AI research community. We look forward to any further engagement.
>
> 1. Case Studies and Empirical Validation | We appreciate the desire for empirical validation and detailed case studies to support the RAD-AI framework. In our response to Reviewer 1, we include examples of AI implementation by organisations including the ABC, CommBank, and Google, each of which can help illustrate the potential for practical application of RAD-AI. While this position paper is not intended as an empirical study, these examples ground our framework and show how RAD-AI thinking is already influencing real world decision making. We agree that further formal case studies would be a valuable direction for future work building on this foundational framing and this is indeed part of our ongoing research.
>
> 2. Main Call-to-Action | The primary call-to-action is an increased uptake of the RAD-AI framework within the ML research community. Our paper offers a shared conceptual grounding and practical bridge between AI researchers, context-specific domain experts and end-users who are expected to adopt AI tools, and policymakers seeking to ensure contextual and societal expectations are addressed and maintained. RAD-AI offers a practical and structured articulation of what responsible application means for each of these communities, and a way to ensure AI practitioners are accountable to it. RAD-AI is a framework designed to be context-sensitive rather than one-size-fits-all. It highlights the need for co-development and alignment with legal, cultural and ethical expectations across domains. For example, in Education, the context requires AI researchers to  navigate AI's interaction with minors and support rather than undermine pedagogical goals (Popenici and Kerr 2017, tinyurl.com/zfwet8h3). In the Insurance sector, the context necessarily involves greater commercial objectives than Education, but there are still several legal requirements and ethical sticking points that must be navigated (Mullins et al. 2021, tinyurl.com/3ykjjhjb). RAD-AI enables AI researchers and partners working in these sectors to articulate and negotiate such tensions clearly and proactively.
>
> 3. Distinctiveness of the Proposal | We appreciate that the RAD-AI framework may seem broadly intuitive, and perhaps even obvious, to those at the forefront of Responsible AI discussions. We welcome this, as it suggests that it is an approach that is practical and applicable. Further, we consider that its formalisation provides a necessary and overdue evolution of application-driven ML, responsible research and innovation for the broader ML research community - particularly for the next generation of multi-disciplinary researchers. RAD-AI's framework provides a shared reference across disciplines and formalises intuition into a set of replicable, testable processes, building on insights from a broad range of fields including ML, social sciences, law and technology, and contextually-relevant domain areas.
>
> Just as Codes of Conduct or Best Practices often codify intuitive or obvious elements to ensure clarity and consistency across a community, RAD-AI offers a structured lens for incorporating essential components of the research and development process, and for making visible the tensions and assumptions that often remain implicit in applied AI work. The novelty is not so much in the elemental components of the approach, but in its overall composition and direction - both of which we argue would be transformative for AI methods and applied science.
>
> Crucially, RAD-AI frames responsible, application-driven AI practice not just as a technical challenge, but as a knowledge challenge. For instance, while ML researchers may define “knowledge” in terms of obtaining accurate predictions from a model, domain experts often define “knowledge” through contextual understanding, stakeholder needs, and lived experience. RAD-AI provides a way to expose and navigate these epistemological tensions, which are rarely surfaced by other frameworks.
>
> By creating a language and structure for these discussions, RAD-AI aims to empower a wide range of experts through transdisciplinary teams to meaningfully contribute to the development and critique of AI systems in ways that would not be possible with ML expertise alone.
>
> Finally, recent public concerns and governance interventions in areas such as autonomous vehicles (Stilgoe and Cohen, 2021; Hemesath and Tepe, 2023) and drone delivery services (Zenz and Powles, 2024) highlight that we still lack the robust, repeatable processes needed to ensure RAI practices are consistently implemented across diverse contexts. Meaningful progress requires ongoing dialogue and deeper transdisciplinary engagement.

---

### Official Review · Reviewer_GXsb · 2025-03-17

**Significance:** 2
**Argument Clarity:** 2
**Rating:** 3
**Confidence:** 3

**Questions:**

RAD-AI requires AI researchers to engage deeply with societal needs, which might slow down traditional AI development cycles. How should researchers and institutions balance the need for responsible, participatory research with the fast-paced nature of AI innovation?

The paper emphasizes context-specific methods as key to RAD-AI while also advocating for scalability and efficacy. However, these goals can sometimes conflict—how do you propose balancing the need for tailored AI solutions with the scalability required for widespread adoption?

**Discussion Potential:**

2

**Paper Summary:**

This paper advocates for Responsible, Application-Driven (RAD) AI Research, arguing that AI research should be guided by real-world applications and ethical considerations rather than purely technical advancements. It critiques traditional Responsible AI (RAI) frameworks for being vague and detached from the practical needs of society. The authors propose a three-stage approach to RAD-AI: building transdisciplinary teams, developing context-specific methods, testing, and sustaining efficacy.

**Position:**

Yes

**Position In Title:**

Yes

**Related Work:**

2

**Strengths And Weaknesses:**

Strengths:

The paper presents a well-structured argument advocating for Responsible, Application-Driven (RAD) AI research.
The discussion challenges technological determinism and emphasizes the importance of community participation in AI development.

Weakness:
The discussion on scalability and industry adoption could be expanded, particularly on how RAD-AI can be integrated into corporate and academic research settings.

**Support:**

2

---

> ### Author Rebuttal · Authors · 2025-04-01
>
> Thank you for your insightful feedback. We appreciate the opportunity to address your concerns regarding potential tensions in the RAD-AI approach, particularly around scalability and industry adoption; and balancing responsible, participatory research with fast-paced AI innovation.
>
> 1. Scalability and Industry Adoption | We agree that context-specific methods can at times be in tension with scalability and efficacy.  However, we maintain that this is a virtue of the RAD-AI approach, particularly in contrast to current approaches to Responsible AI (RAI) implementation. As stated by Rolnick et al. (2024), ``methods originally tailored for a specific problem have proven useful to a variety of seemingly dissimilar problems.'' If scalability were our sole focus, we might miss opportunities to develop innovative methods that benefit ML research broadly.
>
> We also welcome the invitation to elaborate on integrating RAD-AI into corporate and academic research. Scaling RAD-AI effectively involves developing a community of practice that spans academia and industry towards a common implementation approach. Institutional champions are crucial in both sectors; senior leadership plays a pivotal role in fostering responsible practices by shaping organisational culture, establishing ethical frameworks, and orienting organisations to their many stakeholders (London, 2024, dx.doi.org/10.2139/ssrn.4736880). RAD-AI can be integrated through interdisciplinary research, collaborative projects across academia and industry, and courses on RAD-AI. Initiatives are starting to emerge such as the University of California Berkeley's new College of Computing, Data Science, and Society, the Responsible AI Research Centre (tinyurl.com/mupzxdbk), and two multi-sectoral initiatives in the Netherlands: Ethical, Legal and Societal Aspects Labs and Innovation Center for AI Labs.
>
> Implicit in the feedback is an understandable realism about the investment needed for deep engagement with societal needs. Again, we see it as a strength that our approach integrates this engagement in a planned manner. While industry, research institutions, and the public sector broadly agree on the need for ethical and responsible AI development, implementation approaches diverge (Jobin et al, 2019, doi.org/10.1038/s42256-019-0088-2). We use 3 examples to show how high-level principles must be flexible to translate into relevant considerations at a context-specific level. First, Australia's national broadcaster, the ABC, is researching ways to align the Australian AI Ethics Principles with ABC Charter obligations and media sector expectations, reflecting RAD-AI principles (Seneque et al. 2024). The ABC's methodological questions and path for implementation can be contrasted with another Australian organisation, the Commonwealth Bank (Commbank), which is pushing the boundaries on AI techniques to identify and reduce technology-assisted abuse (github.com/h2oai/aitd). While the actions of both entities are underpinned by the same high-level Australian AI Ethics Principles, their translation to context-specific solutions requires a nuanced, tailored process. Third, Google's RAI Context in AI Research team has been demonstrating significant elements of the RAD-AI approach in the development of ethical healthsheets through contextualised, community-driven-AI methods, made scalable by collaborating with an international consortium and grassroots organisations (tinyurl.com/GoogleCAIR).
>
> 2. Participatory Research at the Speed of Tech | RAD-AI should complement, not replace, existing fast-paced AI work. Current AI innovation is necessary but incomplete; RAD-AI provides a crucial counterbalance. This mirrors the debate between the "publish or perish" mindset and "slow science," as popularised by Isabelle Stengers (2018). Both are important, but RAD-AI sits on the "slow science" side of the ledger, and requires building trust and co-creating knowledge with communities, aligning innovations with societal needs from the start.
>
> By conducting engaged research from the start, solutions developed through RAD-AI are designed for immediate uptake, avoiding the scramble currently seen by organisations to understand and utilise innovations post-development. Centring people and problems ensures that innovation is specifically designed for applied adoption. While RAD-AI may progress more slowly than other AI innovation styles, this rich process aims to avoid misaligned solutions, avoiding counterproductive setbacks and accelerating long-term impact.
>
> We are firmly of the position that this need not be an either-or situation. One way to balance fast-paced AI innovation is through iterative sprints of co-creation. This approach allows for rapid development cycles while maintaining the integrity of responsible, participatory research. Moving forward, there is need to interrogate the AI development lifecycle to determine where RAD-AI can be complemented with fast-paced AI innovation sprints.

---

### Decision · Program_Chairs · 2025-04-30

**Decision:**

Accept (poster)

**Comment:**

Reviewers felt that the paper presented a well-articulated position with strong evidence and a compelling framework.